Gene signatures with predictive and prognostic survival values in human osteosarcoma

Qiu Zhongpeng 1
Du Xinhui 1
Chen Kai 1
Dai Yi 1
Wang Sibo 1
Xiao Jun 2
Li Gang ligang_shzu@shzu.edu.cn 1
1 Trauma Department of Orthopedics, First Affiliated Hospital, School of Medicine, Shihezi University , Shihezi , Xinjiang , China
2 School of Medicine, Shihezi University , Shihezi , Xinjiang , China
Altun Zekiye
Electronic publication date: 2021 Jan 15
Publication date: 2021
Volume: 9
Electronic Location ID: e10633
Received 2020 Jul 21; Accepted 2020 Dec 1
Copyright: ©2021 Qiu et al.
Copyright year: 2021
Copyright holder: Qiu et al.
License: This is an open access article distributed under the terms of the Creative Commons Attribution License, which permits unrestricted use, distribution, reproduction and adaptation in any medium and for any purpose provided that it is properly attributed. For attribution, the original author(s), title, publication source (PeerJ) and either DOI or URL of the article must be cited.
License URL: https://creativecommons.org/licenses/by/4.0/

Keywords: Osteosarcoma, Gene signature, Prognosis, Survival rate, Coexpression analysis

Funding: The authors received no funding for this work.

==============================
Osteosarcoma is a common malignancy seen mainly in children and adolescents. The disease is characterized by poor overall prognosis and lower survival due to a lack of predictive markers. Many gene signatures with diagnostic, prognostic, and predictive values were evaluated to achieve better clinical outcomes. Two public data series, GSE21257 and UCSC Xena, were used to identify the minimum number of robust genes needed for a predictive signature to guide prognosis of patients with osteosarcoma. The lasso regression algorithm was used to analyze sequencing data from TCGA-TARGET, and methods such as Cox regression analysis, risk factor scoring, receiving operating curve, KMplot prognosis analysis, and nomogram were used to characterize the prognostic predictive power of the identified genes. Their utility was assessed using the GEO osteosarcoma dataset. Finally, the functional enrichment analysis of the identified genes was performed. A total of twenty-gene signatures were found to have a good prognostic value for predicting patient survival. Gene ontology analysis showed that the key genes related to osteosarcoma were categorized as peptide–antigen binding, clathrin-coated endocytic vesicle membrane, peptide binding, and MHC class II protein complex. The osteosarcoma related genes in these modules were significantly enriched in the processes of antigen processing and presentation, phagocytosis, cell adhesion molecules, Staphylococcus aureus infection. Twenty gene signatures were identified related to osteosarcoma, which would be helpful for predicting prognosis of patients with OS. Further, these signatures can be used to determine the subtypes of osteosarcoma.

Introduction

Osteosarcoma (OS) is a common primary malignancy seen mainly in children and adolescents (Yang et al., 2018). Its annual incidence ranges from two to three in 1 million people (De Azevedo et al., 2020), and it accounts for 40.51% of primary bone malignancies (Li et al., 2020). The incidence of OS is higher in males than in females (Damron, Ward & Stewart, 2007), and it accounts for 15% of all extracranial tumors in the 10–19-year-old age group) (Nie & Peng, 2018). It is considered the third most common cancer in adolescence (Zhang, Lan & Lin, 2018). OS treatment is aggressive and combines neoadjuvant chemotherapy, extensive surgical resection, and additional postoperative adjuvant chemotherapy. The five-year survival rate of non-metastatic patients is 65%–70% (Siegel, Miller & Jemal, 2018). Regarding patients with distant metastases, the 5-year survival rate is only 15%–30% (Whelan & Davis, 2018). Further, relapse rate remains high at approximately 35% (Yu et al., 2014). Metastasis remains the main cause of death in patients with OS. Many prognostic factors are reported to be related to OS: miR-195, miR-21, TGF-β, MMP-9, HIF-1, APE1, and COX2. However, use of these factors has not improved survival rate related to OS (Zamborsky et al., 2019). Therefore, clarifying the molecular mechanisms underlying the occurrence and progression of OS and exploring the potential diagnostic and therapeutic targets are of great importance for the diagnosis and treatment of OS. Many biomarkers have been evaluated (Table 1) (Wan-Ibrahim et al., 2015), but none have been approved by the Food and Drug Administration for clinical settings (Wan-Ibrahim et al., 2015). This lack may be due to research deficiencies, unacceptable heterogeneity, or absence of effective evaluation. Biomarkers for patients with OS, particularly those with metastases, are urgently needed for early diagnosis and establishment of treatment goals. Serum biomarkers are used for predicting prognosis of other cancers but are rarely characterized in OS (Zamborsky et al., 2019). An obvious need in OS is effective biomarkers for characterizing disease progression and associated prognosis.

Table 1 Potential biomarkers for osteosarcoma.

Study approach	Type of biomarker	Candidate biomarker	Regulation	Sample	Technique	Reference	
Genomics	Diagnostic	C7orf24	↑	Tissue; cell lines	RT-PCR; qRT-PCR; Western blot (WB); siRNA transfection	Uejima et al. (2011)	
Diagnostic And Predictive	miRNA signatures	−	Tissue; cell lines	qRT-PCR	Gougelet et al. (2011)	
Predictive	Multigene classifier	−	Tissue	cDNA microarray; qRT-PCR	Man et al. (2005)	
Predictive	Multigene classifier	−	Tissue	cDNA microarray; RT-PC	Mintz et al. (2005)	
Predictive And Prognosis	miRNA signatures	−	Tissue	miRNA microarray; qRT-PCR; immunohistochemistry (IHC)	Jones et al. (2012)	
Predictive And Prognosis	miR-21	↑	Serum	qRT-PCR	Yuan et al. (2012)	
Prognosis	Tenascin-C	↓	Cell lines	cDNA microarray; RT-PCR; WB; IHC	Xiong et al. (2009)	
Prognostic	miRNA-214	↑	Tissue	qRT-PCR	Wang et al. (2014)	
Proteomics	Diagnostic	Serum amyloid A (SAA)	↑	Plasma	SELDI-TOF MS; WB	Li et al. (2010)	
Diagnostic	Ezrin (EZR); a crystallin β chain (CRYAB)	↑	Tissue	2D-DIGE; LC-ESI-MS/MS; RT-PCR for mRNA; tissue microarray and IHC	Folio et al. (2009)	
Diagnostic	Cytochrome C1 (CYC-1)	↑	Serum; cell lines	SELDI-TOF MS; Gene microarray (cell lines)	Li et al. (2009)	
Diagnostic	Zinc finger protein 133 (ZNF 133); tubulin-a1c (TUBA1C)	↑	Tissue	2-DE; MALDI-TOF MS; WB; IHC	Li et al. (2010)	
Diagnostic	Protein NDRG 1	↑	Plasma membrane from cell line and tissue	2-DE; LC-ESI-MS/MS; IHC; WB	Hua et al. (2011)	
Diagnostic	Gelsolin	↓	Serum	2D-DIGE; MALDIHTOF; WB; ELISA	Jin et al. (2012)	
Predictive And Prognostic	SAA	↑	Serum	2D-DIGE; MALDI-TOF MS; WB; ELISA	Jin et al. (2007)	
Predictive	Peroxiredoxin 2 (PRDX2)	↑(poor prognosis)	Tissue	2D-DIGE; LC-ESI-MS/MS; WB	Kikuta et al. (2010)	
Predictive	SAA; transthyretin (TTR)	↑(poor prognosis)	Plasma	SELDI-TOF MS; WB	Li et al. (2011)	

One study reported that CDC20 and its downstream substrates, secure, cyclin A2 and cyclin B2 are good prognostic factors for OS (Wu et al., 2019). Savage et al. (2013) suggested that two loci in the GRM4 gene at 6p21.3 and in the gene, desert, at 2p25.2, These two loci warrant further exploration to uncover the biological mechanisms underlying susceptibility to osteosarcoma.The study addressed a single gene and did not take into account interactions among molecules that regulate tumorigenesis. Three candidate genes (ALOX5AP, CD74 and FCGR2A) were found. Their expression levels in lung and lymph nodes were higher than levels in matched cancer tissues, and they may be expressed in microenvironments (Li et al., 2020). Some limitations exist in these studies. First, accuracy cannot be guaranteed with only one dataset because of an expected high false-positive rate. Further, using a single high-throughput analysis method (only sequencing or chip data), results obtained will be biased. Second, a patient’s sample data are too limited. Finally, clinical information is incomplete.

Identifying the minimum number of robust genes needed to produce a predictive signature for prognosis for patients with OS was the objective of this study. The lasso regression algorithm was used to analyze sequencing data from TCGA-TARGET, and Cox regression analysis, risk factor score, receiving operating curve (ROC), KMplot prognosis analysis, nomogram and other methods were used to assess genes for their predictive power. Next, the accuracy and predictive power of twenty-gene identified in this process were assessed using the GEO OS dataset. Finally, we performed functional enrichment analysis on these twenty-gene.

Material and Methods

Data collection and preprocessing

Training set: The TARGET-OS RNA-sequencing dataset (presented as fragments per kilobase million, FPKM), corresponding clinical characteristics and prognosis information were downloaded from UCSC Xena (Goldman et al., 2019) (https://xena.ucsc.edu/). Patients with expression profiles but no prognostic information and clinical characteristics were excluded. Finally, 84 patients with OS were included in a training set. FPKM data were converted to TPM data and annotated using gencode.v22.annotation.gene.probeMap.

Validation set: The gene expression data GSE21257 (Buddingh et al., 2011) (GPL570 (HG-U133_Plus_2) Affymetrix Human Genome U133 Plus 2.0 Array) for 53 patients with OS were downloaded from the GEO database, and corrected and annotated with R software.

Construction of gene signatures

A linear regression multiple regression model was developed for the underlying expression levels of genes for prognostic risk scores. The method chosen by lasso Cox was 10-fold cross-validation. According to the median cutoff value (the cutoff value refers to the content before the brackets of the HR value in each dataset) of the risk score, patients with OS were divided into high-risk and low-risk groups. Model prediction efficiency using the training set was evaluated by Kaplan–Meier log-rank test, time-dependent ROC curve analysis, Cox regression analysis, and risk factor score for validation and test sets. A nomogram was constructed using Iasso’s guidelines.

Weighted correlation network analysis of genes

Based on the variance of gene expression in TARGET-OS data, the top 5000 genes were selected for WGCNA (Langfelder & Horvath, 2008). This analysis proceeded as: check for outliers in all samples, construct sample tree with hclust, and remove outliers according to cut height. To explore the correlation between expression data and clinical phenotypes, the sample tree and characteristic heat map were visualized. Subsequently, the strength of associations between pairs of nodes of the adjacency matrix aij was calculated as: sij = |cor (xi, xj)| aij = Sijβ. xi and xj are genes i and j. The vector of expression value, sij, indicates the strength of Pearson’s correlation coefficient between genes i and j. The aij coding network connects genes i and j. β value is a soft threshold (power value). Further, the Scale-Free Topology Fit Index (scale-free R2) range from 0 to 1 is used to determine the scale-free topological model. Selecting a set of soft threshold powers (range: 1 to 20) assists in calculating scale-free topological model fitting. The soft threshold of β = 7 was used to define the adjacency matrix. The corresponding scale-free R2 value is 0.87, suggesting a satisfactory scale-free topology model.

In coexpression networks, the highest absolute association genes were clustered into the same module to generate a clustering dendrogram. Relationships between clinical traits and risk scores were analyzed by Pearson correlation and results were visualized by heat map analysis. Genes in the module were analyzed for gene ontology (GO) and KEGG pathway enrichment. Moreover, Cytoscape (Smoot et al., 2010) (version 3.7.2) was used to visualize the weighted coexpression network. gene ontology (GO) analysis, which includes annotation of biological processes (BPs), molecular functions (MFs), and cellular components (CCs), serves as a major bioinformatics tool to annotate genes and analyze the biological processes of these genes. Huimei Wang et al. analyzed the biological classification of DEGs, showing that changes in BPs of DEGs were significantly enriched in positive regulation of associated cell response, by GO analysis (Wang et al., 2018).

Functional enrichment analysis

Clusterprofiler (Yu et al., 2012) was used to study modules related to biological function for determining functional and pathway enrichment. A multiple testing correction was performed using hypergeometric test functions and the Benjamini–Hochberg method. The GOplot (Walter, Sánchez-Cabo & Ricote, 2015) package was used to visualize the enrichment analysis.

Statistical analysis

Statistical analysis was conducted in R software (version 3.6.1) with the following packages: “glmnet” (Friedman, Hastie & Tibshirani, 2009), “survivalROC” (Heagerty, Lumley & Pepe, 2000), “WGCNA,” and “clusterProfiler.” All the statistical tests were two sided, and P-values of < 0.05 were considered statistically significant.

Results

Construction of genes classifier for OS

Prognostically significant genes for OS in TARGET-OS data were analyzed and a total of 1151 genes were significantly associated with poor prognosis. These 1151 candidate genes were included in the lasso prognostic classifier for further screening and model construction. A twenty-gene classifier for the OS was developed (Figs. 1A and 1B). The gene information in the model is shown in Table 1. Patients were divided into a high-risk group (n = 42) and low-risk group (n = 42) based on risk scores. The median risk score was set as the cutoff (the cutoff value refers to the content before the brackets of the HR value in each dataset). Table 2 shows clinical characteristics of patients with OS in the training set according to their high risk and low risk scores. The Kaplan–Meier log-rank test suggested a significant difference between high-risk and low-risk groups in the training set (P < 0.001; Fig. 1C). In the time-dependent analysis of the ROC curve, AUCs for OS in the first, third, and fifth years were 0.94, 0.98, and 0.97, respectively (Fig. 1D).

Figure 1 Construction of comprehensive prognostic classifier based on the training set.

A-10-fold cross-validation for tuning parameter selection in the LASSO model for OS.B-LASSO coefficient profiles of 20 prognostic genes for OS.C-Kaplan–Meier overall survival with a low or high risk of death in the training dataset.D-Time dependent ROC curves at 1, 3 and 5 years for OS OS overall survival.

Table 2 Twenty genes significantly related to the overall survival in the training set underlying the LASSO model.

Ensemble ID	Gene	Chromosome location	Coefficient of lasso model	HR	p value	
ENSG00000103274.9	NUBP1	chr16:10743786-10769351:(+)	−0.2216	0.905 (0.86, 0.953)	0.000	
ENSG00000163219.10	ARHGAP25	chr2:68679601-68826833:(+)	−0.0503	0.795 (0.685, 0.924)	0.003	
ENSG00000077420.14	APBB1IP	chr10:26438203-26567803:(+)	−0.0429	0.924 (0.88, 0.97)	0.001	
ENSG00000162517.11	PEF1	chr1:31629862-31644896:(-)	−0.0230	0.979 (0.966, 0.992)	0.002	
ENSG00000102226.8	USP11	chrX:47232690-47248328:(+)	−0.0123	0.982 (0.97, 0.994)	0.004	
ENSG00000179163.11	FUCA1	chr1:23845077-23868294:(-)	−0.0104	0.969 (0.949, 0.99)	0.003	
ENSG00000189171.12	S100A13	chr1:153618787-153634092:(-)	0.0019	1.007 (1.004, 1.011)	0.000	
ENSG00000132535.17	DLG4	chr17:7189890-7219702:(-)	0.0088	1.064 (1.021, 1.109)	0.003	
ENSG00000158315.9	RHBDL2	chr1:38885807-38941799:(-)	0.0144	1.01 (1.006, 1.013)	0.000	
ENSG00000176171.10	BNIP3	chr10:131966455-131981931:(-)	0.0170	1.009 (1.004, 1.014)	0.000	
ENSG00000167549.17	CORO6	chr17:29614756-29622907:(-)	0.0425	1.054 (1.021, 1.088)	0.001	
ENSG00000125337.15	KIF25	chr6:167996241-168045089:(+)	0.0570	1.148 (1.076, 1.224)	0.000	
ENSG00000179262.8	RAD23A	chr19:12945855-12953642:(+)	0.0741	1.005 (1.002, 1.008)	0.000	
ENSG00000147378.10	FATE1	chrX:151716035-151723194:(+)	0.0862	2.217 (1.518, 3.238)	0.000	
ENSG00000113739.9	STC2	chr5:173314713-173329503:(-)	0.0895	1.016 (1.005, 1.027)	0.004	
ENSG00000197467.12	COL13A1	chr10:69801931-69964275:(+)	0.0910	1.021 (1.013, 1.029)	0.000	
ENSG00000017483.13	SLC38A5	chrX:48458537-48470256:(-)	0.1003	1.008 (1.003, 1.014)	0.002	
ENSG00000136997.13	MYC	chr8:127735434-127741434:(+)	0.1635	1.003 (1.002, 1.005)	0.000	
ENSG00000138028.13	CGREF1	chr2:27098889-27119115:(-)	0.2105	1.015 (1.009, 1.021)	0.000	
ENSG00000241563.3	CORT	chr1:10449719-10451902:(+)	0.2504	1.026 (1.016, 1.036)	0.000	
Notes.

HR Hazard Ratio

CI Confidence Interval

Identification of a prognostic risk score model based on the training set

Based on univariate regression analysis (Cox’s proportional hazard model), forest plots of selected genes with p-value and hazard ratios are shown in Fig. 2A. HR, Hazard Rate in Fig. 2A, shows that the p-value of selected genes was less than 0.05, showing the statistical difference, which cannot be ignored. Only when 95% CI conclude 1, it could be proved that these selected genes signatures were not significantly associated with the prognosis of patients with OS. Although the extremum of HR is very close to 1, they show that these selected genes might be associated with the prognosis of patients with OS. Risk scores and survival status of patients with OS are shown in Figs. 2B and 2C, respectively. These results indicate that patients with high risk scores have poor outcomes compared with patients with low risk scores (Fig. 2D). Using these data, a nomogram combining the classifier with clinicopathological features to predict the survival probability of patients with different risk scores was prepared (Fig. 3A). The calibration chart demonstrated that predicted three-year and five-year survival rates were very close to observed ratios (Fig. 3B).

Figure 2 Multivariate Cox regression analysis was performed for the selected genes, and the risk scores, survival status and risk heat maps of the 20 prognostic genes were distributed in the training set.

(A) Hazard ratio distribution for selected 20 key genes.distribution of risk scores, (B) overall survival of 84 patients, and heatmap of 20 genes in prognostic classifier in the training set. The black dotted line suggested the median cutoff divividing patients into low-risk and high-risk groups.

Figure 3 Nomograms to predict 1-year, 3- year and 5-year survival probability in osteosarcoma.

(A) Total points were obtained by incorporated the corresponding points of recurrence, metastasis, risk score on the point scale. The total points were then converted into specific 1 year, 3- yearand 5-year associated survival probabilities. (B) Calibration plot underlying the nomogram. Dashes show the nomogram-predicted probability for each group, as well as the respective confidence intervals.

Validation of twenty-gene signature for survival prediction in the validation set

Subsequently, the validation set was used to assess the power of the twenty-gene signature in predicting prognosis. OS of the low-risk group was superior to the high-risk group (P < 0.05; Fig. 4A). Also, time-dependent ROC analysis is used to assess the effectiveness of risk models in predicting outcomes. Areas under ROC was 0.81, 0.78, and 0.76 at one, three and five years, respectively, suggesting that the twenty-gene signature displays good predictive power (Fig. 4B). The C-index value is 0.944. The risk score (Fig. 4C) and survival status (Fig. 4D) of patients with OS, and distribution of risk scores of twenty-gene expression profiles are shown in a heat map of 53 patients in the validation set (Fig. 4E). These results suggest that the twenty-gene signature shows good prognostic value for patient survival. This finding is further validated since patients with high risk scores were associated with poorer prognoses compared with patients with low risk scores. Previous studies have investigated gene factors in an attempt to identify new prognostic OS markers (Goh et al., 2019; Guan, Guan & Song, 2020). Hence, I have compared the prognostic gene set for osteosarcoma, Cox univariate and multivariate analysis showed that BACE2, ING2 ALOX5AP, HLA-DMB, HLA-DRA, and SPINT2 were not the independent prognostic factors for osteosarcoma (Table 3). Table 3 shows the determinants of BACE2, ING2 ALOX5AP, HLA-DMB, HLA-DRA, and SPINT2 for all the patients. In the univariate analysis, These gene signatures exhibited a negative correlation with age, gender, specific and primary tumor site, but illustrate a positive correlation with recurrence and risk score. After multiple stepwise analysis, these biomarkers correlated independently with Recurrence (hazard ratio (HR) 5.374, 95% CI [1.183–24.404], p = 0.029) and risk score (HR 9.869, 95% CI [4.663–20.887], p < 0.001).

Figure 4 Kaplan–Meier survival, time-dependent ROC curves, and risk score distribution, the overall survival status of 84 patients, and heat maps of the expression of 20 genes in the low and high risk groups based on the validation set.

(A) Kaplan–Meier curve of osteosarcoma patients with a low or high risk of death.(B) Time dependent ROC curves at 1, 3, and 5 years for OS in the validation set. (C) Distribution of risk scores for genes in the validation set, (D) overall survival of 53 patients, (E) and heat maps of the expression of 20 genes in the low-risk and high-risk groups.

Table 3 Univariate and multivariate Cox regression analyses genes signiture for patients with osteosarcoma in study cohort.

	Univariate	Multivariate	
Characteristics	HR	p.value	HR	p.value	
Gender (Female vs Male)	0.666 (0.317, 1.399)	0.283			
Age (<18 vs ≥18)	0.909 (0.345, 2.393)	0.846			
Recurrence (No vs Yes)	19.348 (4.589, 81.563)	<0.001	5.374 (1.183, 24.404)	0.029	
Primary tumor site (Arm/Hand OR Pelvis vs Leg/Foot)	0.487 (0.168, 1.413)	0.185			
Specific tumor site (Femur vs Tibia or others)	0.642 (0.309, 1.337)	0.236			
riskscore	13.915 (7.067, 27.398)	<0.001	9.869 (4.663, 20.887)	<0.001	

Gene co-expression network analysis

The relationship between risk scores and gene expression profiles was evaluated using the top 5000 genes in the variance filter for WGCNA analysis. The red line (cut height = 8000) was used to remove the abnormal samples in the sample tree. TARGET-40-PAUXOZ-01A was excluded after removing outliers (Fig. 5). The sample dendrogram and trait heat map placed selected samples into different sample clusters that provided clinical trait information (Fig. 6A). Independence and average connectivity of coexpression modules were determined by power (β) and scale R2 value. A series of soft thresholds and corresponding performance power were plotted. The threshold for scale R2 value was set at 0.85. The power value of seven is the threshold that first reaches the scale R2 value, and was chosen as the soft threshold to construct and identify coexpression modules (Figs. 6B and 6C). The expression matrix was converted to an adjacency matrix, and subsequently into a topology matrix. Based on TOM, genes are clustered based on criteria of mixed dynamic shear trees using average connection–level distance. The minimum number of genes in each module was set at 7. Finally, a total of 11 modules were identified with the WGCNA package (Fig. 6D). Statistics of gene numbers in each module are shown in Table 4. Associations of these modules with clinical characteristics (including sex, age, relapse, OS, metastasis, major cancer sites, specific cancer sites, and risk scores) are shown in Fig. 6E. T module showed the highest correlation with the risk score. Further, the magenta module also was negatively correlated with sex and OS and positively correlated with major cancer sites. GSEA analysis between high-risk and low-risk groups can better illustrate the risk-score related to the biological process. But GSEA analysis between high-risk and low-risk groups only considers the expression level of gene sets. Analysis of WGCNA is based on the clinical relevant factors, which makes it sense, although its risk-score is low.

Figure 5 Sample cluster analysis to identify outliers underlying RNA sequencing data.

The red line suggested the cut-off data filtering in the data preprocessing step c.

Figure 6 Overview of co-expression modules identified from the osteosarcoma RNA-seq dataset based on the training set.

(A) Sample dendrogram and trait heat map underlying gene expression data and clinical information.(B) Scale independence and (C) average network connectivity with different soft threshold powers (). Select a soft threshold power of 7 to achieve maximum model fit. (D) Cluster dendrogram of the co-expression modules of the 20 genes identified. Each differentially expressed gene represents a leaf, and each of the six modules consists of a branch. The lower panel displays the colors assigned to each module. Note that the gray blocks indicate unassigned genes.(E) Weighted correlation of module features between the identified modules and clinical features and corresponding P-values. The color scale on the right represents the correlation of module features from -1 (blue) to 1 (red). Green represents perfect negative correlation, and red represents perfect positive correlation.

Table 4 Relationship between risk score of the 20-marker-based prognostic classifier with os and clinical characteristics in the training set.

Characters	Level	Low risk	High risk	P value	
n		42	42		
Gender	Female (%)	16 ( 38.1)	21 (50.0)	0.379	
	Male (%)	26 ( 61.9)	21 (50.0)		
Age	<18 (%)	32 ( 76.2)	34 (81.0)	0.791	
	≥18 (%)	10 ( 23.8)	8 (19.0)		
Overall survival	Alived (%)	42 (100.0)	13 (31.0)	<0.001	
	Dead (%)	0 ( 0.0)	29 (69.0)		
Recurrence	No (%)	32 ( 76.2)	13 (31.0)	<0.001	
	Yes (%)	10 ( 23.8)	29 (69.0)		
Metastasis	No (%)	37 ( 88.1)	26 (61.9)	0.011	
	Yes (%)	5 ( 11.9)	16 (38.1)		
Primary tumor site	Arm/Hand OR Pelvis (%)	3 ( 7.1)	5 (11.9)	0.713	
	Leg/Foot (%)	39 ( 92.9)	37 (88.1)		
Specific tumor site	Femur (%)	13 ( 31.0)	25 (59.5)	0.015	
	Tibia or others (%)	29 ( 69.0)	17 (40.5)		

Functional enrichment analysis in the gene coexpression network of the magenta module

GO and KEGG enrichment analysis was used to characterize biological functions of genes in the magenta module as they related to risk scores. GO analysis showed that key genes related to OS were mainly enriched in peptide–antigen binding, clathrin-coated endocytic vesicle membrane, peptide binding, and MHC class II protein complex (Fig. 7A). KEGG analysis showed that the key genes related to OS were mainly enriched in antigen processing and presentation, phagosome, cell adhesion molecules, Staphylococcus aureus infection (Fig. 7B). Subsequently, the interaction network for enrichment pathways in the magenta module was visualized (Fig. 7C). Moreover, an interaction network was constructed to visualize genes in the coexpression magenta module (Fig. 7D).

Figure 7 Functional enrichment analysis and WGCNA weighted network analysis of magenta module.

(A) GOplot indicated the correlation between magenta module genes and their related GO terms. (B) GOplot indicated the correlation between magenta module genes and their related KEGG terms. (C) KEGG pathway analysis of magenta module genes based on clusterprofiler. (D) WGCNA weighted network diagram of magenta module.

Discussion

OS is a disease involving complex interactions among many factors. Overall, OS disrupts cell signaling pathways, causing loss of bone tissue homeostasis (Otoukesh et al., 2018). The urgent need to obtain better clinical results highlights the related need for better diagnostic, prognostic, and predictive biomarkers (Ludwig & Weinstein, 2005). Presently, no specific markers are available for OS diagnosis. To reduce mortality and increase limb salvage, biomarkers are needed for early identification of disease (Smida et al., 2017). Possible genetic biomarkers to address this need were identified in the present study. Lasso regression screened in twenty-gene in a training set. These genes have certain prognostic value in time ROC, risk factor, and Kaplan–Meier plots. Results were verified with a validation set.

One current study showed that inhibition of BNIP3 expression by baicalein treatment could inhibit cell apoptosis (Ye et al., 2015). Moreover, the MYC gene is also reported to be amplified in a subset of OS (Ladanyi et al., 1993). MYC promotes the proliferation of OS cells through the autophagy pathway (Mo et al., 2019). Cross-species genomics identified DLG2 as a tumor suppressor in OS (Shao et al., 2019). Further, depletion of KIF25 leads to the formation of actin stress fibers, which may be due to the changes of Rho signaling observed before microtubule destabilization (Wittmann & Waterman-Storer, 2001). In addition to the above studies of genes related to OS, relationships between other genes screened in this study and malignancies have been reported in the literature but have not been reported in OS. A lack of related research reports to help assess the potential of these genes as targets for the treatment and diagnosis of OS currently exits.

The current study also explored the relationship between risk scores and gene expression profiles using WGCNA analysis. Important genes related to OS were enriched in 11 different modules. Results showed that pathways related to inflammation and immunity were primarily enriched in the turquoise module. The gray and turquoise modules share the most pathways among all pairwise comparisons. Genes in these modules may play similar roles in OS.

LASSO is a method of shrinking and variable selection linear regression model. The purpose of LASSO regression is to obtain a subset of the predictors to minimize the prediction errors of the quantitative response variables. The lasso does this by constraining the model parameters, making the regression coefficients of some variables approach 0. “Coxnet” fits a lasso penalty, and its adaptive forms, such as adaptive lasso. Moreover, it treats the number of non-zero coefficients as another tuning parameter and simultaneously selects with the regularization parameter “lambda”. And Xiong et al. (2020) established a 13 gene-based survival score for prognostic prediction of Lung adenocarcinoma. They filtrated the relevant gene by LASSO and identified the wide applicability of these genes.

Previous studies report that CXCR3 may be an independent prognostic risk factor, suggesting a possible benefit of immunotherapy for OS (Tang et al., 2019). The susceptibility and severity of OS may also be related to functional polymorphism of inflammatory genes (Oliveira et al., 2007). Another study has shown that increased expression of MIF indicates an increased risk of metastasis, and MIF is related to angiogenesis and cell infiltration of OS. MIF can be used as a prognostic marker of OS and a potential therapeutic target (Kim et al., 2008). Therefore, clarifying molecular mechanisms of OS may facilitate the identification of novel therapeutic and prognostic targets. Limitations to the study exist. First, although these twenty-gene were identified to have certain prognostic value for OS, all data analyzed in the study were retrieved from the online databases. Thus, further experimental evidence, such as real-time PCR, western blot, immunohistochemistry assays, is required to fully elucidate the role of 20-gene signatures. Second, to determine the diagnostic accuracy of gene associations, a larger sample size would be useful for additional internal validation. Third, assessment of all clinically relevant influencing factors for OS was not included, and more clinical information and PFS-related data are needed. Finally, differentially expressed genes were not evaluated and nor was the need to add a normal control group and joint verification of multiple tumor sites.In conclusion, the current study proposes a 20-gene signature for diagnostic and prognostic purposes for OS. The twenty-gene signature is independently related to prognostic parameters of OS classification. Also, the signature is a good classifier for different subtypes of patients with OS. This signature may provide a new perspective on the prognosis of OS. The biological functions and pathways enriched in specific modules will be beneficial to the development of new therapeutic methods for the treatment of OS.

Supplemental Information

Supplemental Information 1 Supplementary tables

Click here for additional data file.

Supplemental Information 2 Patients’ information and characteristics

Click here for additional data file.

Supplemental Information 3 Magenta-module GO kegg

Click here for additional data file.

Supplemental Information 4 Magenta-module GO kegg and gene ontology analysis of that are involved with the genes in the magenta co-expression module

Click here for additional data file.

Additional Information and Declarations

Competing Interests

Author Contributions

Data Availability

The authors declare there are no competing interests.

Zhongpeng Qiu and Gang Li conceived and designed the experiments, performed the experiments, analyzed the data, authored or reviewed drafts of the paper, and approved the final draft.

Xinhui Du and Kai Chen conceived and designed the experiments, performed the experiments, analyzed the data, prepared figures and/or tables, and approved the final draft.

Yi Dai and Sibo Wang analyzed the data, prepared figures and/or tables, and approved the final draft.

Jun Xiao conceived and designed the experiments, authored or reviewed drafts of the paper, and approved the final draft.

The following information was supplied regarding data availability:

The raw measurements are available in the Supplemental Files.

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
