# Peer review of "Gene signatures with predictive and prognostic survival values in human osteosarcoma"

_PeerJ, doi:10.7717/peerj.10633_

## Round 0.1 · original submission · Major Revisions

Your work is requiring some revisions and then it can be evaluated for publication in PeerJ.

·

Basic reporting

1. The English of the manuscript is clear.
2. The authors should compare and describe the prognostic gene set for osteosarcoma (X Guan et al., 2020. Cancer Cell Int; Goh TS et al., 2019. J Cell Physiol).
3. The data in the manuscript is proper and raw data is accessible because they used open databases.

Experimental design

1. The manuscript is included in aims and scope of peerj.
2. The study design is reasonable and sounds clear. For the readers, please add study flow as a figure, it will help to understand the study.
3 and 4. They performed several statistical methods and they described the methods in detail.

Validity of the findings

1. They developed the prognostic gene set using LASSO. LASSO is one of the most popular and statistically advanced methods. However, they did not compare the previous prognostic gene sets.
2. The statistical methods of the manuscript sound clear.
3. The conclusions and limitations in the manuscript are well stated.

Additional comments

The authors developed and validated prognostic gene set for osteosarcoma using open databases. The manuscript is interesting, but, there are some scientific issues to address.

1. The authors should compare and describe the prognostic gene set for osteosarcoma (X Guan et al., 2020. Cancer Cell Int; Goh TS et al., 2019. J Cell Physiol).

2. They developed the prognostic gene set using LASSO. LASSO is one of the most popular and statistically advanced methods. There are several grouped variable selection methods including LASSO in the coxnet R package (Goh TS et al). In the 'user service' in easysurv (Pak K et al., 2020. JMIR, https://easysurv.net), the authors can perform Elastic net and Net methods after uploading the data. Please perform other grouped variable selection methods and describe them in the manuscript.

3. Please describe the cutoff values in each dataset.

4. Please perform the univariate and multivariate cox regression of risk score as a continuous variable (not categorical variable) in each dataset.

5. (Table 3) Please add the general patients' information and characteristics according to the risk score in the training and validation set.

Reviewer 2 ·

Basic reporting

no comment

Experimental design

no comment

Validity of the findings

no comment

Additional comments

The author did an interesting study, which is of scientific and clinical significance to some extent. The role of gene expression by rna-seq was studied by using some simple but effective methods. I think the author's idea is very clever and commendable, The authors used public data, which is a reasonable use of resources since many high-throughput sequencing data have not been fully developed. By using the bioinformatics, the authors point out a direction of prognostic prediction for OS, which is very instructive for other relevant researchers. The authors not only reveal the key genes in the OS prognostic genes, but also explore the influence of different biomarkers. This is helpful to reveal the molecular mechanism of OS and to develop new drugs. I think it is acceptable for publishing, but there some problems needs to be revised before publication.

In table 1, the author stressed the potential biomarkers for osteosarcoma, this was well summarized. But the meaning for this should be compared with the authors’ signature. Is the authors’ signature more accuracy than these biomarkers or not. The authors should clearly point out the results.

There is no experimental validation in this study, and the author should discuss the limitation of this part at least.

There were a lot of signatures for OS and they should be discussed in this study, for examples Liu et al (PMID: 31146489) developed a pseudogene signature for OS prognosis and the author should explain or discuss the difference or connection of the these studies and clearly cited corresponding manuscript.

How to explain the extremum of HR in fig2a, they are very close to 1 and this may mean they are not so important?

20 genes consisted a signature, that was too complex, I think the author can use step-wise cox regression to simplify the signature, because if 5 genes can get a AUC=0.95, why we need to test 20 genes to get AUC=0.95?.

Please pay attention fig1d and fig4b was not time-dependent ROC curve, it just ROC curve. The authors should clearly recognize these two different analysis. If the authors are interested to time-dependent ROC curve, it would be much better to add this analysis in this study.

The author did a lot of GO analysis, please explain the meaning or usage of them.

The module related to risk-score has a low correlation coefficient and this means these genes may related to risk-score little. I think gsea analysis between high-risk and low-risk group can better illustrated the risk-score related biological process.

---

## Round 0.2 · accepted · Accept

This manuscript contains new bioinformatic data about osteosarcoma according. It suggests a 20 genes signature for diagnostic and prognostic purposes for OS. These bioinformatic data can be validated by further analysis.

·

Basic reporting

no comment

Experimental design

no comment

Validity of the findings

no comment

Additional comments

The authors performed the revision properly.